# Magnesium(II) Porphyrazine with Thiophenylmethylene Groups-Synthesis, Electrochemical Characterization, UV–Visible Titration with Palladium Ions, and Density Functional Theory Calculations

**DOI:** 10.3390/molecules29153610

**Published:** 2024-07-30

**Authors:** Wojciech Szczolko, Kyrylo Chornovolenko, Jacek Kujawski, Zbigniew Dutkiewicz, Tomasz Koczorowski

**Affiliations:** 1Chair and Department of Chemical Technology of Drugs, Poznan University of Medical Sciences, Rokietnicka 3, 60-806 Poznan, Poland; stud7data@gmail.com (K.C.); zdutkie@ump.edu.pl (Z.D.); 2Chair and Department of Organic Chemistry, Poznan University of Medical Sciences, Rokietnicka 3, 60-806 Poznan, Poland; jacekkuj@ump.edu.pl

**Keywords:** macrocyclization, aminporphyrazine, palladium, electrochemistry, titration

## Abstract

The presented studies aimed to evaluate the peripheral coordinating properties of a novel porphyrinoid family representative preceded by its synthesis for potential sensing purposes. Two synthetic pathways were employed to a obtain maleonitrile derivative, further used as a starting material in the cyclotetramerization reaction. In the first one, DAMN was used in sequential double-reductive alkylation with 2-thiophene-carboxyaldehyde and sodium borohydride. In the second, DAMN was used in a one-pot reaction with 2-thiophene-carboxyaldehyde in the presence of a 5-ethyl-2-methylpyridine borane complex in methanol and acetic acid. Following the Linstead approach, the cyclization reaction led to a novel symmetrical magnesium(II) octaaminoporphyrazine with methyl(2-thiophenylmethylene) substituents. The macrocycle’s electrochemical properties were assessed by cyclic and differential pulse voltammetries revealing one reduction and two oxidation peak potentials. The additional spectroelectrochemical measurements showed formation of a cationic form of the macrocycle at an applied potential of 0.6 V. The coordinating properties due to the palladium ion of novel porphyrazines were measured with the use of titration combined with UV–vis spectrometry. The titration of Pd^2+^ revealed the good sensing activity of porphyrazine in the range of 0.1 to 5 palladium molar equivalents. In addition, Pd^2+^ ions coordination was also assessed by electrochemical studies, indicating the peak potential shift of 0.1 V in the presence of metal cations. DFT calculations showed the good agreement between theoretical and experimental data in the UV–vis and ^1^H NMR studies.

## 1. Introduction

Porphyrinoids, which are tetrapyrrole macrocyclic complexes, encompass a significant category of organic compounds. Among them, there are two distinct synthetically derived analogs: porphyrazines (Pzs) and phthalocyanines (Pcs). These analogs exhibit a distinct structural feature—*meso* nitrogen bridges that connect pyrrolyl or isoindolyl substituents, respectively. Their physicochemical, optical, catalytic, and electrochemical properties are tunable due to commonly employed methods: introducing peripheral substituents and exchanging metal cations within the macrocyclic core [1,2,3]. Porphyrazines can be categorized into sulfanyl and amino derivatives, depending on the starting materials for the cyclotetramerization reaction.

Porphyrazines have been extensively explored as chelating agents and sensing materials due to their extended π-electron system, the presence of various metal cations inside the core, and heterocyclic substituents in the periphery. Some amino-porphyrazines have also been analyzed as molecular scaffolds [4]. Among various assessments, magnetostructural studies of metal core interactions in spin-coupled multimetallic systems are essential since they provide a basis for efforts to produce high-spin molecules and molecular arrays based on metal-ion exchange coupling [5]. Porphyrazines are suggested to be excellent options for this application because their outer regions may be easily functionalized with heteroatoms to coordinate additional metal ions. This provides an opportunity to look into spin interaction among the localized moments of several metal ions via the porphyrazine ring [5]. Following this path, porphyrazines based on the 1,10-phenanthroline nucleus were obtained, in which the π-conjugated system of the porphyrazine ring was extended, fusing an aromatic phenanthroline unit to its periphery; they displayed efficient metal coordination and were subsequently converted into a collection of Ru(II) complexes [6,7].

Moreover, several other examples of porphyrazines containing sulfur at their periphery have been investigated concerning their chelation properties. Pzs derivatives containing soft S donor atoms have significantly influenced solid-state interactions. For instance, dithia-crowned porphyrazine exhibits advantages over its analogues, wherein two S atoms in each crown unit directly conjugate with the 18 π-electron core of the porphyrazine [8,9]. In another example, asymmetrical porphyrazines carrying 2,5-di(2-thienyl)-1H-pyrrole and dimethylamino substituents on the periphery prepared by Goslinski and White [10] reveal high coordinating potential against metal ions.

Moreover, porphyrinoids, like phthalocyanines and porphyrazines, have been recognized as efficient electrocatalysts for sensing various biologically important compounds, including H_2_O_2_, ascorbic acid, and L-glutathione [11,12]. Possibly due to their similarities to naturally occurring porphyrinic oxygenase enzymes, many studies have focused on their application as oxidation catalysts. For instance, iron(II) thioporphyrazines, which mimic the structure of iron-containing cytochrome P450 enzymes, have been investigated by Deng and colleagues for application in the oxidative degradation of organic pollutants [13,14]. The ability of a majority of porphyrazines to promote the generation of singlet oxygen may be the most significant application of these compounds as catalysts, rather than their role as ligands in metal-mediated reactions [15]. The singlet oxygen produced upon irradiation may be used in a variety of oxidation processes, such as [4 + 2] cycloaddition reactions resulting in endoperoxides and ene reactions to yield allylic hydroperoxides [16].

In terms of the electrocatalytic abilities of some porphyrinoids and their potential applicability as electrode-modifying materials, these compounds have been already widely investigated in the field of electrochemical sensors [11]. However, in the case of certain phthalocyanines, the reduction in the electron transfer rate can occur due to their low electrical conductivity [17,18]. Moreover, some phthalocyanines directly embedded on electrode surfaces showcased inadequate surface coverage and stability during measurements [12]. To overcome these obstacles, porphyrazines, with their smaller core size and higher stability, emerge as promising alternative electrocatalysts, especially when the adsorption of macrocycles on carbon-based nanomaterials has been implemented. Some of the recent studies have utilized various porphyrazines with diverse heteroaromatic peripheries in fabricating electrochemical sensors for biologically active substances after deposition on carbon-based nanomaterials [19].

Many researchers have also paid attention to the electrochemical properties of porphyrazine derivatives [20,21]. Belviso et al. assessed them to yield new fluorinated porphyrazines with liquid-crystalline and spectro- and electrochemical characteristics suited for possible use as electronic materials [22]. The sulfanyl Pzs bearing nitrophenoxy and isophthaloxyalkyl substituents recently synthesized by Hassani et al.’s group, revealed plenty of interesting photochemical and electrochemical properties [23]. What is more, aza-porphyrinoids have already been studied for their potential combination with ferrocene [24,25]. Such combinations were discovered to have similarities to photosynthesis-active regions, making them potential composites for electronics and molecular electrochemical sensors.

Given that the possible sensing properties of porphyrazines towards metal cations still need further investigation, in the present study, we have synthesized a novel magnesium aminoporphyrazine possessing eight thiophenylmethylene groups to address this demand. UV–vis spectroscopy and cyclic voltammetry assessed the macrocycle’s coordination abilities towards palladium ions. Moreover, we have performed electrochemical and spectroelectrochemical analyses of targeted porphyrazine and compared the NMR spectroscopy results with the quantum-chemical calculations.

## 2. Results and Discussion

### 2.1. Synthesis and Physicochemical Characterization

The synthesis leading to the targeted macrocyclic compound was preceded by the fabrication of the starting material. This was led through a series of maleonitrile derivatives (Figure 1). The pre-starting precursor **5** was obtained by two synthetic pathways. In the first one, diaminomaleonitrile (DAMN) (**1**) was used in sequential double-reductive alkylation with 2-thiophene-carboxyaldehyde and sodium borohydride leading to 2,3-bis-2-thiophenylmethylene)amino]-2(*Z*)-butene-1,4-dinitrile (**5**) [1,26]. In the second one, DAMN was used in a one-pot reaction with 2-thiophene-carboxyaldehyde in the presence of the 5-ethyl-2-methylpyridine borane complex in methanol and acetic acid [27]. This reaction led to a mixture of **4** and **5** with good yield. Next, the alkylation reaction of **5** using dimethyl sulfate in the presence of sodium hydride in a temperature range from −20 to −10 °C [28] led to the alkylated 2,3-bis[methyl(thiophenylmethylene)amin]-2(*Z*)-butene-1,4-dinitrile (**6**).

Linstead macrocyclization of the tetraalkylated DAMN derivative **6** with magnesium butanolate as a base in 1-butanol led to a symmetrical porphyrazine **7** (Figure 2). All of the obtained new compounds were characterized by MS ESI (for maleonitrile derivatives) or MADLI TOFF (for pz **7**), UV–vis, ^1^H, and ^13^C NMR. In addition, the CHN elemental analyses were performed for all small molecules. The electronic spectrum of **7** in dichloromethane revealed characteristic peaks for porphyrinoid-based dyes: a B-band peak at 353 nm, resulting from the absorbance of individual pyrrolyl moieties in the macrocyclic ring and thiophene groups at the periphery, and a sharp Q-band at 710 nm from the π-electron-conjugated system of the macrocyclic ring (see Supporting materials). No additional signals were detected. In the ^1^H NMR, five signals were detected. In the aliphatic region, two singlets at 3.61 ppm and 5.68 ppm were assigned to eight methyl CH_3_- substituents next to peripheral nitrogen atoms and methylene –CH_2_- linkers between the peripheral nitrogen and thiophene groups, respectively. In the case of the aromatic region, one multiplet signal at 6.89–6.91 ppm, alongside two doublets at 7.04–7.05 ppm and 7.29–7.30 ppm, were originating from aromatic protons of thiophene rings, comprising a total of 24 H. In the ^13^C NMR spectrum, signals at 41.9 ppm and 54.1 ppm were assigned to methyl group carbon and methylene carbon, respectively. Other signals present on the spectrum represented aromatic pyrrolyl and thiophene moieties. The number of signals detected, and their integrity, confirmed the chemical structure of the synthesized compound **7**. The ^1^H and ^13^C NMR spectra can be seen in Appendix A.

### 2.2. Titration of Porphyrazine 7 with Palladium Ions

After the physicochemical characterization, we subjected porphyrazine **7** to UV–vis titration studies with palladium ions, where significant changes in the spectra were observed, probably due to a formation of the heterobimetallic porphyrazine product (carrying the exocyclic methyl(thiophenylmethylene)amin-PdCl_2_-bonded fragment). This study aimed to evaluate the possible chelation properties of the obtained macrocyclic compound provided by thiophene substituents at the periphery. Titration with 0.1 eq of Pd(II) resulted in the formation of a less intensive, broader, and blue-shifted Q-band by approx. 50 nm compared to the starting spectrum. Q-band intensity at the Pd(II)-ligand ratio starting from 0.1 eq began to decrease with the increase in the concentration of Pd ions to the ratio of 1 (Figure 1). Moreover, at concentrations corresponding to the amount of porphyrazine and palladium ions 1:2–1:4, it was observed that the Q band becomes more and more symmetrical. At a ratio of 1:5, a new band was observed at 741 nm. The titration was continued until a ratio of 1:10 was reached. At the ratio of 8, a new bond was observed at 842 nm, the intensity of which increased to a ratio of 10. The formation of a new band at 741 nm and 842 nm can be explained by the coordination of palladium ions by nitrogen atoms in the *meso* position of porphyrazine [29,30].

The possible structures of the heterobimetallic porphyrazine **7** complex with palladium(II) ions are presented in Figure 2. Based on the UV–vis data, at Pd^2+^ ratios ranging from 0.1 to 4 eq, the palladium(II) ions are coordinated to sulfur atoms from thiophene peripheral substituents (**A**). At higher ratios, from 5 to 10 eq, the coordination may additionally occur at the *meso* nitrogen atoms in the macrocyclic ring (**B**).

Alongside UV–vis titration studies, we performed fluorescence measurements of selected mixtures of porphyrazine-Pd^2+^ complexes. As shown in Figure 3, emission intensity increased while palladium ions were added with no peak shifting. Such a phenomenon can be explained by changes in the electronic structure of formatting heterobimetallic complexes, which may change the energy of the electronic levels and cause an increase in the fluorescence quantum yield.

### 2.3. Electrochemical and Spectroelectrochemical Studies

Porphyrazine **7** with peripheral thiophenylmethlene substituents was subjected to electrochemical and spectroelectrochemical characterization to assess its redox activity and susceptibility to cationic or anionic species formation. The experiments were carried out in dichloromethane containing 0.1 M tetrabutylammonium perchlorate as a supporting electrolyte due to the insolubility of the macrocycle in aqueous solutions. The electrochemical measurements were performed in a classic three-electrode system using a glassy carbon working electrode. The obtained peak potentials were adjusted to ferrocene/ferrocenium redox pair potential as ferrocene was used as the internal standard. The cyclic and differential pulse voltammograms of **7** were taken in a −1.5 V to 1.2 V electrochemical window. CV experiments were performed at five scan rates from 25 mV/s to 200 mV/s for better assessment of the diffusion and adsorption currents. Within the electrochemical window, three redox processes were recorded—one reduction at −1.76 V and two oxidations at −0.30 and 0.27 V, respectively (Figure 4). None of them were assigned as reversible processes, as the calculated ΔE_1/2_ were higher than 100 mV in each case.

The differential pulse voltammogram revealed some additional redox processes at a positive potential related to the aggregation of porphyrazine in dichloromethane on the surface of the electrode while scanning (Figure 5). What is more, the high oxidation peak currents observed at more positive potentials (>0.5 V) might result from the presence of sulfur atoms at the periphery of the macrocycles, which are highly susceptible to oxidation due to two lone electron pairs. Such a phenomenon was observed earlier in the case of symmetrical magnesium(II) sulfanyl porphyrazines [31,32].

The electrolysis of **7** in DCM/0.1M TBAP solution was performed to investigate the ability of porphyrazine to form anionic or cationic species. In these spectroelectrochemical tests, Pt gauze was used as a working electrode. The formation of derivatives was monitored by the changes in the UV–vis spectrum in the range of 400–1000 nm upon applied potential within 2 min. The measurements were taken in a 1 mm path length quartz cuvette. The utilized applied potential ranged from −1.5 to 1.0 V. However, the results showed that only at 0.6 V did the significant changes in the UV–vis spectra occur (Figure 6). At this specific potential, the decrease in Q-band absorption at approx. 710 nm was observed with the simultaneous appearance of a new red-shifted band at approx. 850 nm. Due to the presence of an electrochemical inactive metal cation inside the macrocyclic core, this could indicate the formation of a cationic specie of porphyrazine **7,** as the spectra changes occurred at a positive potential.

### 2.4. Electrocatalytic Studies

To evaluate the chelating abilities of the synthesized porphyrazine **7**, an electrocatalytic study was conducted using PdCl_2_ as an analyte. Prior to the experiment, the glassy carbon working electrode needed to be modified by embedding compound **7** onto the surface of the GC electrode, which was previously deposited on multi-walled carbon nanotubes. The resulting GC/MWCNT/**7** electrode was then subjected to electropolymerization in oxygen-free PBS to obtain stable CV voltammograms. For the experiment, the modified working electrode was immersed in a 1 mM PBS solution of palladium chloride with the addition of 100 µL of acetone to enhance analyte solubility without impacting the deposited porphyrazine. CV scans were conducted at 25 mV/s and compared to those obtained using a bare GC electrode (Figure 7). The results indicated a shift of the Pd^2+^ reduction peak potential towards negative values by approximately 0.1 V when the modified electrode was used, suggesting the possible chelation of palladium ions to the methyl(thiophenylmethylene)amino periphery of porphyrazine **7**.

### 2.5. Computational Studies

To match the experimental (Figure 1) and theoretical UV–vis spectra of analyte **7**, we optimized the molecules’ geometry and applied the linear response time-dependent DFT (TDDFT) method for the calculations. The vertical excited states were calculated for each optimized rotamer I (Appendix A) of compound **7** at the B3LYP/6-311++G(2d,3p) level of theory in dichloromethane (CPCM solvation model).

The contours of the highest occupied molecular (HOMO) and lowest occupied molecular (LUMO) orbitals for **7** are presented in Figure 8. They were visualized based on the checkpoint file (.chk) generated during the TD-DFT computations. The HOMO and LUMO orbitals are located mainly over the pyrrole ring and the -C=N- linker. The HOMO–LUMO gap calculated for 7 at the B3LYP/6-311++G(d,p) level is 2.1233 eV, corresponding to an electron transition from spinorbital 350 to spinorbital 351. It can be assigned to the calculated first excitation state at 682.63 nm. The first excited state for compound **7** relates mainly to the band corresponding to a HOMO→LUMO transition (the coefficient is 0.63708, calculated energy is 1.8163 eV, and oscillator strength f = 0.1979; data are taken from the output file). In this case, the HOMO−LUMO contribution relative to the first excited state, calculated as the duplicated coefficient square, is 81%. The calculated maximum of absorption is related to the calculated seventh excitation state at 537.98 nm (the coefficient is 0.63444, calculated energy is 2.3046 eV, and oscillator strength f = 0.2049; data are taken from the output file) corresponding to an electron transition from spinorbital 347 to spinorbital 351 (HOMO-3 → LUMO). In this case, the HOMO−LUMO contribution is 81%. The above discussion shows that the DFT method can satisfactorily explain the observations taken from the experimental UV–vis spectra of the analyzed compound.

Next, in our paper we focused our attention on the computation of the ^1^H NMR spectrum of **7**. Although the compounds have nitrogen atoms within their structure, the low abundance of the ^15^N isotope in comparison with the ^1^H isotope, as well as the significant broadening of signals due to the large quadrupole moment of N, renders the nitrogen NMR spectroscopy impractical. ^1^H NMR spectroscopy is broadly used in confirming the identity and purity of small-molecule organic compounds. Considering the low utility of ^13^C NMR spectroscopy for the investigation of interactions of small-molecule organic compounds as well as the low natural abundance of ^13^C, we decided to use ^1^H NMR for the studies depicted in our paper. The theoretical calculations of the chemical shifts for **7** were executed for all energy minima found during the alteration of dihedral angles connected with the rotation of the C-N(CH_3_)-CH_2_ fragment and thiophene moiety, respectively (the cartesian coordinates of conformers I−VI are given in Appendix A). The calculated proton chemical shifts for the optimized conformers are compared with the experimental ones and those given in Table 1. The relative error for proton C is almost 8% due to the steric hindrance and the dynamic zig-zag-like type of coupling with the p orbitals of the nitrogen atom within the C-N(CH_3_)-CH_2_ moiety connected to the central porphyrazine ring in the solvation sphere. A large error is associated with the position of protons C and B—the relative error is about 8 and 21%, respectively, for the structure at global minimum. Likely explanations for this inaccuracy are the steric and solvation effects that determine the spatial arrangement of the N(CH_3_)-CH_2_ linker.

## 3. Materials and Methods

### 3.1. General Experimental Methods

All reactions were conducted in oven-dried glassware under argon. All solvents were rotary evaporated at or below 50 °C. The reaction temperatures reported refer to external bath temperatures. Methanol, tetrahydrofuran, and dichloromethane were distilled. Other solvents and all reagents were obtained from commercial suppliers and used without further purification unless otherwise stated. Melting points were obtained on a “Stuart” Bibby apparatus and are uncorrected. Dry flash column chromatography was carried out on Merck silica gel 60, particle size 40–63 µm and Fluka silica gel 90. Thin-layer chromatography (TLC) was performed on silica gel Merck Kieselgel 60 F_254_ plates and visualized with UV (λ_max_ 254 or 365 nm). The UV–vis spectra were recorded on a Hitachi UV/VIS U-1900 spectrometer; λ_max_ (log ε), nm. Elemental analyses and mass spectra (ES) were recorded by the Advanced Chemical Equipment and Instrumentation Facility at the Faculty of Chemistry, Adam Mickiewicz University in Poznan, whereas HRMS (MALDI TOF) spectra were detected at the Wielkopolska Centre for Advanced Technologies in Poznan. ^1^H NMR, ^13^C NMR spectra were recorded using a Bruker 400 and 500 spectrometer. Chemical shifts (δ) are quoted in parts per million (ppm) and are referred to as residual solvent peaks. Coupling constants (J) are quoted in Hertz (Hz). The abbreviations s, b, d, t, and m refer to singlet, broad, doublet, triplet, and multiplet, respectively.

### 3.2. Synthesis and Characterization of New Compounds



**2-Amino-3-[(2-thiophenylmethylene)amino]-2-butene-1,4-dinitrile (2)**



2-Thiophenecarboxaldehyde (1.8 mL, 20 mmol), diaminomaleonitrile (**1**) (2.16 g, 20 mmol) and trifluoroacetic acid (3 drops) in MeOH (30 mL) were stirred for 2 min when a yellow solid precipitated out of the solution. The solid was filtrated and washed with Et_2_O-hexanes (1:1) (20 mL) to imine **2** (3.45 g, 85%) as a yellow solid: mp 196 °C dec; R_f_ 0.26 (CH_2_Cl_2_); ^1^H NMR (500 MHz, DMSO-*d*_6_) δ_H_ 8.45 (s, 1H), 7.84–7.83(d, ^3^J = 4.8 Hz, 1H), 7.75–7.74 (d, ^3^J = 4.8 Hz, 1H), 7.69 (bs, 2H), 7.20–7.18 (t, ^3^J = 4.5 Hz 1H); ^13^C NMR (125 MHz, DMSO-*d*_6_) δ_C_ 149.7, 147.0, 142.3, 141.9, 134.5, 134.3, 132.8, 132.6, 131.2, 130.7, 129.2, 128.8, 126.4, 114.9, 114.3, 113.6, 112.7, 104.5, 103.4; MS (ESI) m/z 203 [M+H]^+^, 225 [M+Na]^+^; Anal. Calc. for C_9_H_6_N_4_S: C, 53.45; H, 2.99; N, 27.70. Found: C, 53.63; H, 3.32; N, 27.84.
**2-Amino-3-[(2-thiophenylmethyl)amino]-2-butene-1,4-dinitrile (3)**

NaBH_4_ (0.277 g, 7.5 mmol) was slowly added to a rapidly stirring solution of imine **2** (1.011 g, 5 mmol) in MeOH (25 mL). After the addition was complete, the solution was stirred for 15 min and poured into ice–H_2_O (200 mL). A solid was formed, filtrated, and washed with H_2_O (50 mL) to give derivative **3** (0.821 g, 80%) as a dark tan solid: mp 149–151 °C, R_f_ 0.23 (CH_2_Cl_2_); ^1^H NMR (500 MHz, DMSO-*d*_6_) δ_H_ 7.46–7.45 (m, 1H), 7.01–7.00 (m, 2H), 5.75 (s, 2H), 5.7–5.67 (t, ^3^J = 6.0 Hz, 1H), 4.40–4.38 (d, ^3^J = 6.2 Hz, 2H); ^13^C NMR (125 MHz, DMSO-*d*_6_) δ_C_ 143.1, 127.4, 126.0, 126.0, 116.9, 116.1, 110.0, 107.1, 44.4; MS (ESI) m/z 205 [M+H]^+^; 227 [M+Na]^+^; Anal. Calc. for C_9_H_8_N_4_S: C, 52.92; H, 3.95; N, 27.43. Found: C, 53.03; H, 3.72; N, 27.24.
**2-[(2-thiophenylmethylene)amino]-3-[(2-thiophenylmethyl)amino]-2-butene-1,4-dinitrile (4)**

2-Thiophenecarboxaldehyde (0.8 mL, 8.7 mmol), maleonitrile (**2**) (1.5 g, 7.3 mmol) and trifluoroacetic acid (3 drops) in MeOH (25 mL) were stirred for 2 min when a yellow solid precipitated out of the solution. The solid was filtrated and washed with Et_2_O-hexanes (1:1) (20 mL) to give imine **4** (0.85 g, 39%) as a yellow solid: mp 147–149 °C; R_f_ 0.42 (CH_2_Cl_2_); ^1^H NMR 400 MHz, DMSO-*d*_6_) δ_H_ 8.48 (s, 1H), 8.42 (bs, 1H), 7.87–7.86 (d, ^3^J = 4.9 Hz, 1H), 7.78–7.77 (d, ^3^J = 3.5 Hz 1H), 7.49–7.47 (m, 1H), 7.21–7.19 (m, 1H), 7.08–7.07 (d, ^3^J = 2.5 Hz, 1H), 7.03–6.98 (m, 1H), 4.79 (s, 2H); ^13^C NMR (100 MHz, DMSO-*d*_6_) δ_C_ 149.9, 147.3, 142.2, 141.7, 141.3, 141.3, 141.0, 134.3, 132.6, 132.5, 131.1, 130.8, 128.6, 128.3, 127.1, 127.0, 126.9, 126.3, 126.2, 126.0, 125.9, 125.6, 125.6, 115.1, 113.6, 112.3, 112.3, 111.6, 109.8, 104.9, 103.9, 44.1, 43.9; MS (ESI) m/z 299 [M+H]^+^; Anal. Calc. for C_14_H_10_N_4_S_2_: C, 56.35; H, 3.38; N, 18.78. Found: C, 56.51; H, 3.28; N, 18.84.
**2,3-Bis-[(2-thiophenylmethyl)amino]-2(*Z*)-butene-1,4-dinitrile (5)**

Two experiments have been performed:

(1) NaBH_4_ (0.111 g, 3 mmol) was slowly added to a rapidly stirring solution of imine **2** (0.600 g, 2 mmol) in MeOH (20 mL). After the addition was complete, the solution was stirred for 15 min and poured into ice–H_2_O (200 mL). A solid was formed, filtrated, and washed with H_2_O (50 mL) to give derivative **5** (0.394 g, 69%) as a dark tan solid: mp 91–93 °C, R_f_ 0.35 (CH_2_Cl_2_); ^1^H NMR (400 MHz, DMSO-*d*_6_) δ_H_ 7.47–7.46 (dd, ^3^J = 1.4 Hz, 2H), 7.01–6.97 (m, 4H), 6.13–6.10 (t, ^3^J = 6,5 Hz, 2H), 4.44–4.43 (d, ^3^J = 6.2 Hz 2H); ^13^C NMR (100 MHz, DMSO-*d*_6_) δ_C_ 142.7, 127.4, 126.2, 126.1, 115.6, 110.2, 110.1, 44.4, 44.3; MS (ESI) m/z 301 [M+H]^+^; 323 [M+Na]^+^; Anal. Calc. for C_14_H_12_N_4_S_2_: C, 55.97; H, 4.03; N, 18.65. Found: C, 55.69; H, 3.92; N, 18.34.

(2) Diaminomalonitrile **1** (0.432 g, 4 mmol), 2-thiophenecarboxaldehyde (0.734 mL, 8.0 mmol), and acetic acid (2 mL) in MeOH (10 mL) were stirred, 5-ethyl-2-methylpyridine borane complex (0.6 mL, 4 mmol) was added dropwise for 30 min. After 2 h of stirring at room temperature, hydrochloric acid (1 mL) was added and the mixture was stirred for 2 h at 50 °C. Then, the mixture was evaporated to dry residue and chromatographed to give yellow solid **5** (0.400 g, 33%) and **4** (0.198 g, 16%)
**2,3-Bis-[methyl(2-thiophenylmethyl)amino]-2(*Z*)-butene-1,4-dinitrile (6)**

NaH (60% dispersion in mineral oil; 0.032 g, 0.8 mmol) was added to a cooled (−15 °C) THF, a solution of **5** (0.100 g, 0.33 mmol) in THF (2 mL) was added dropwise. After 30 min, Me_2_SO_4_ (0.076 mL, 0.8 mmol) in THF (1 mL) was dropwise added to the solution for 20 min and the yellow mixture was stirred at room temperature for 24 h. The solution was poured into ice–H_2_O (50 mL) to give **6** (0.082 g, 82%) as yellow crystalline solid: mp 158–161 °C, R_f_ 0.4 (CH_2_Cl_2_); ^1^H NMR (400 MHz, DMSO-*d*_6_) δ_H_ 7.50–7.49 (m, 2H), 7.02–7.00 (m, 4H), 4.41 (s, 4H), 2.77 (s, 6H); ^13^C NMR (100 MHz, DMSO-*d*_6_) δ_C_ 138.3, 127.6, 126.9, 126.5, 116.4, 114.6, 50.9, 41.0 (hidden); MS (ESI): m/z 329 [M+H]^+^, 351 [M+Na]^+^; Anal. Calc. for C_16_H_16_N_4_S_2_: C, 58.51; H, 4.91; N, 17.06. Found: C, 58.79; H, 4.82; N, 17.32.
**[2,3,7,8,12,13,17,18-Octakis-[methyl(2-thiophenylmethyl)amino]-porphyrazinato]magnesium (II) (7)**

Mg turnings (0.080 mg, 3.5 mmol), a crystal of I_2_, and 1-butanol (20 mL) were heated under reflux for 4 h. After the mixture was cooled to room temperature, maleodinitrile **6** (577 mg, 1.76 mmol) was added and the solution was heated under reflux for a further 20 h. After being allowed to cool to room temperature, the dark violet mixture was filtrated through Celite and evaporated with toluene (2 × 50 mL). Chromatography (CH_2_Cl_2_, ethyl acetate:hexane 3:7) gave porphyrazine **7** (0.154 g, 26%) as a dark violet thin film: mp 143 °C, R_f_ 0.83 (ethyl acetate:hexane 5:7); UV–vis (CH_2_Cl_2_) λ_max_ nm (log ε) 353 (4.34), 710 (3.89); ^1^H NMR (400 MHz; DMSO-*d*_6_): δ_H_, ppm 7.30–7.29 (m, 8H), 7.05–7.04 (m, 8H), 6.91–6.89 (m, 8H), 5.68 (s, 16H), 3.61 (s, 24H); ^13^C NMR (100 MHz, DMSO-*d*_6_) δ_C_ 151.5, 142.7, 137.7, 126.4, 125.9, 125.1, 54.1,.41.9. MS (MALDI TOF): m/z 1378 [M+H]^+^.

### 3.3. General Procedure for UV–Vis Titrations of Porphyrazine ***7***

A solution of known concentration of porphyrazine **7** in dichloromethane was subjected to UV–vis titrations with solutions of varying concentrations (0, 0.01, 0.1, 0.5, 1, 2, 3, 4, 5, 6, 7, 8, 10 molar equiv.) of PdCl_2_(C_6_H_5_CN)_2_ in dichloromethane. Blank UV–vis spectra (in the absence of metal salt) were run for **7** to determine any solvent effect (of which there were none). Blank UV–vis spectra (in the absence of porphyrazine **7**) were also run for the metal salt to make sure that there was no significant absorbance in the window of interest (300–1000 nm).

### 3.4. Electrochemical Measurements

Electrochemical experiments were performed using the Metrohm Autolab PGSTAT128N potentiostat attached to a PC for data acquisition and storage, driven by GPES (Eco Chemie, Utrecht, The Netherlands) and Metrohm Nova 2.1.4 software. Organic measurements were made in dichloromethane with a glassy carbon (GC) working electrode (3 mm, area = 0.02 cm^2^), Ag wire as a pseudo-reference electrode, and a platinum wire as a counter electrode. Prior to the electrochemical experiments, the GC electrode was polished with an aqueous 50 nm Al_2_O_3_ slurry (provided by Sigma-Aldrich) on a polishing cloth, followed by subsequent washing in an ultrasonic bath with water/acetone mixture for 10 min in order to remove organic and inorganic impurities. Ferrocene was used as an internal standard. Before each measurement, a glass cell (volume 10 mL) containing dichloromethane with a supporting electrolyte (0.1 M TBAP) was deoxygenated by purging nitrogen gas. All electrochemical experiments were carried out at ambient laboratory temperature. DCM and TBAP were purchased from Sigma-Aldrich.

### 3.5. Spectroelectrochemical Measurements

Spectroelectrochemical studies were performed with the use of the Ocean Optics USB 2000+ diode array spectrophotometer and Metrohm Autolab PGSTAT128N potentiostat. Measurements were made in a 1 mm path length quartz cuvette with the use of a Pt gauze working electrode, a Ag/AgCl as a reference, and a platinum wire as an auxiliary electrode in 0.1 M TBAP solution in deoxygenated dichloromethane. Absorbance spectra were recorded in the range of 400–1000 nm within 2 min (every 10 s) during application of proper overpotential.

### 3.6. Modification of Working Electrode and Electrocatalytic Studies

At the beginning, the GC electrode (BASi) was polished with an aqueous 50 nm Al_2_O_3_ slurry (Sigma-Aldrich, Sofia, Bulgaria) on a diamond polishing cloth, followed by subsequent washing in an ultrasonic bath with a water/acetone mixture (1:1, *v*/*v*) for 10 min in order to remove inorganic impurities. Then, the working electrode was immersed in phosphate buffer saline (PBS) and scanned from −0.6 V to 0.4 V to obtain a stable cyclic voltammogram. Next, the surface modification of the working electrode was obtained by the drop-dry method. Following the procedure, 1 mg of porphyrazine **7** was dissolved in 0.5 mL of DMF (TCI Europe, Zwijndrecht, Belgium). Then, the solution of **7** and 0.5 mL of multi-walled carbon nanotubes (MWCNT, Sigma-Aldrich, Burlington, MA, USA) dispersed in DMF (2 mg/mL) were mixed and stirred at room temperature for 60 min leading to a MWCNT/**7** hybrid material in DMF (1 mg/mL). Subsequently, 2 µL of the obtained MWCNT/porphyrazine suspension in DMF was dropped on the surface of the GC electrode and dried in an oven (120 °C) to form a film. In the last step, electropolymerisation was carried out. For this purpose, cycling voltammetry in oxygen-free PBS was applied for 30 cycles between 0.4 and −0.6 V at 100 mV/s (CV scans available in the Appendix A). The obtained electrode GC/MWCNT/**7** was finally rinsed with deionized water in order to remove any traces of impurities. Then, it was used in the electrocatalytic studies in oxygen-free PBS with a platinum wire as the auxiliary electrode and a Ag/AgCl reference. Palladium chloride (TCI Europe) was used as an analyte in a concentration of 1 mM in PBS with addition of 100 µL of acetone (Sigma Aldrich) to improve the PdCl_2_ solubility. The CV scans were recorded in the range 0.4 to −0.6 V at 25 mV/s.

### 3.7. Computational Studies

Quantum-chemical calculations were preceded by searching the conformational space of porphyrazine **7** and generating conformers of this molecule. A three-dimensional model of the demetalated porphyrazine 7 was built in the Chemcraft version 1.8 program, and the atomic coordinates of the model were saved in the XYZ format. The chemical structure of **7** was then subjected to energy minimization in Discovery Studio 3.5 [33] with the use of MMFF94 force field and the adopted basis Newton–Raphson method (RMS gradient tolerance was equal to 0.1 kcal × mol^−1^). The energy-minimized molecule was used as a starting structure for conformational search by applying the ‘Generate Conformations’ protocol with the ‘Boltzmann jump’ algorithm for stochastically searching for conformations. The Boltzmann jump method uses the Metropolis selection criterion to accept or reject random changes in torsional angles. For the studied molecule, this algorithm ensured very small deviations in the planarity of the porphyrazine core, and rotations about single bonds have led to conformations with changed torsion angles describing the orientation of the substituents relative to the core. The Boltzmann temperature was set to 300 K, and conformations were created within the 50 kcal × mol^−1^ energy threshold. After completing the conformational search, 30 distinct conformations satisfying searching conditions were saved. Additionally, three symmetrical structures were built manually. One with C4 symmetry with all substituents on the same side of the porphyrazine core plane, the second with D2, and the third with C2 symmetry. The final structures for quantum chemical calculations were prepared by replacing two hydrogen atoms with a magnesium ion in the center of the porphyrazine core.

The theoretical calculations were executed using Gaussian G16C.01 suite code [34]. The structures of the different conformers of compound **7** were optimized using the density functional theory (DFT) [35] with the B3LYP [36] functional and standard basis 6–31G(d,p) [37]. The solvent effect on the geometry of **7** was determined using a conductor-like polarizable continuum (CPCM) model of dichloromethane (UV–vis) or DMSO (NMR) [38]. The vibrational frequencies and thermodynamic properties were calculated applying the ideal gas, rigid rotor, and harmonic oscillator approximations [39]. The energy minimum was confirmed by the frequency calculation for all conformers. The initial structures of **7** were designed by a change in the C-N(CH_3_)-CH_2_-C_4_H_3_S dihedral angle and rotation of the thiophenyl ring. The NMR parameters were calculated using the coupled perturbed density functional theory (CP-DFT) [40] method with B3LYP functional. This functional theory has been found to be in good agreement with experimental data [41]. For UV–vis calculations, we applied the TD-DFT method [42], CPCM solvation model, the linear response (LR) approach, and dichloromethane as a solvent (B3LYP/6-311++G(2d,3p) approximation). The calculations were performed using the NMR dedicated GIAO [43] method for H, C, N, O, S, and Mg atoms. The proton chemical shifts were referenced to the central signal of residual TMS (0.00 ppm). The TMS geometry was determined in a similar manner by theoretical methods. The proton shift for the B3LYP/6-31G(d,p)//B3LYP/GIAO method equaled 31.7396 ppm.

## 4. Conclusions

The tert-substituted DAMN derivative was prepared by two synthetic routes. In the first one, DAMN was used in sequential double-reductive alkylation with 2-thiophene-carboxyaldehyde and sodium borohydride. The second one consisted of a one-pot reaction of condensation DAMN with 2-thiophene-carboxyaldehyde in the presence of the 5-ethyl-2-methylpyridine borane complex in methanol and acetic acid. The tetracyclization reaction following the Linstead procedure led to a novel symmetrical magnesium(II) octaaminoporphyrazine with methyl(2-thiophenylmethylene) substituents. All compounds were characterized using UV–vis, ^1^H and ^13^C NMR, including 2D techniques, as well as MS (ESI or MALDI). Titration of the obtained macrocycle with palladium ions proved that it has good sensory properties towards this ion in the range from 0.1 eq to 5 eq thanks to the sulfur atoms in the thiopene units at the periphery of the macrocycle. Such a good response of the titration studies predisposes pz **7** for further chelation studies including other heavy metals. Cyclic voltamperometry studies performed in DCM/0.1M TBAP solution revealed one reduction peak potential at −1.76 V and two oxidations at −0.3 and 0.27 V, respectively. Due to the presence of sulfur atoms at the periphery of the macrocycle with two lone electron pairs, pz **7** was highly susceptible to the oxidation process, likewise with similar porphyrazines in the literature. An additional DPV test showed the signals originating from the aggregated species of the macrocycle. In the spectroelctrochemical studies, at 0.6 V the formation of the cationic Pz species were observed due to simultaneous appearance of a new red-shifted band at approx. 850 nm. Such a phenomenon occurred as a result of the presence of peripheral heteroatoms—nitrogen and sulfur, predisposing the formation of cationic forms of porphyrazine, detected on UV–vis upon a positive applied potential. Moreover, the electrocatalytic studies using a glassy carbon electrode modified with multi-walled carbon nanotubes/porphyrazine hybrid nanosystems performed in 1mM PBS solution of PdCl_2_ showed a negative shift of 0.1 V in the case of the prepared electrode, comparing to a bare GC. This also indicated the possible chelation and sensing properties of **7** in the electrochemical analysis. At the end, the additional theoretical quantum chemical calculations confirmed the UV–vis and ^1^H NMR experimental data. In the latter one, in the case of locant B (assigned to –CH_2_- group), it revealed an average error of approx. 21% which can be explained by the steric and solvation effects.

Summarizing, the results of the UV–vis titration and electrocatalytic studies indicated the possible applicability of the synthesized porphyrazine derivative with thiophene peripheral substituents in sensing metal cations in both organic solvents and aqueous environment. The implementation of sulfur-containing moieties enabled the formation of bimetallic complexes of poprhyrazines exhibiting significant changes in UV–vis and emission spectra, as well as on CV voltammograms. This is a good prognosis for further experiments on real water or soil samples, where compound **7** can serve as an optical or electrochemical sensor.

## Data Availability

Data are contained within the article and Appendix A; Samples of the compounds **1**–**7** are available from the authors.

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
