# Peer review of "Magnesium(II) Porphyrazine with Thiophenylmethylene Groups-Synthesis, Electrochemical Characterization, UV–Visible Titration with Palladium Ions, and Density Functional Theory Calculations"

_molecules, 2024, doi:10.3390/molecules29153610_

Round 1

Reviewer 1 Report

Comments and Suggestions for Authors

The authors of the article titled "Magnesium(II) porphyrazine with thiophenylmethylene groups - synthesis, electrochemical characterization, UV-vis titration with palladium ions and DFT calculations" presented the synthesis of porphyrazines with aminothiophene substituents and a complex based on it, properties were studied and computational studies were carried out.

Before recommending this paper some modifications and clarifications are necessary.

First, in the introduction the authors provide data on calexarenes. However, these are different structures from porphyrazines. It is necessary to explain why calexarenes can be considered in this context.

Second, there is almost no discussion of the NMR spectra of the ligands in the description of the synthesis. Characteristic signals should be selected and discussed in the discussion section.

Third, the SI copies of the spectra show the presence of impurities for some substances. Can the authors identify the nature of these impurities and make a comment on this matter.

Fourth, it is necessary to comment on the results of UV spectra. Which absorption bands belong to which groups and aromatic systems.

Fifth, the conclusions should reflect the influence of thiophene substituents on the properties of the obtained substances.

Next, how was the purity of the complex with magnesium established?

Author Response

Dear Reviewer,

We appreciate the time you spent on the revision process. We have addressed all the issues you have raised. In the attachment, you can find the PDF file with our answers and the manuscript's changes.

Kind regards,

Tomasz Koczorowski

Poznan University of Medical Sciences

Reviewer 2 Report

Comments and Suggestions for Authors

The authors present their study of Mg(II) porphyrazine with thiophenylethylene in two synthesis routines. The manuscript studied the titration of the synthesized prophyrazine with Pd ions and showed the UV-vis spectra results. Besides, electrochemical and spectroelectrochemical studies and DFT computational results are provided for more extensive investigations. The manuscript can be published after addressing the following comments:

  1. Regarding the synthesis of Scheme 1 in section 2.1 for the upper routine. Is there a specific reason for not treating compound 1 with two or excess equivalence of 2-thiophenecarboxaldehyde? If the mono-substituted precursor rather shows low solubility or low reactivity towards the second step, would other solvent systems or heating help?

  2. For the titration part, the results are somehow not like precise titration. If I do not miss any important discussion, what is the principle of titration? Is it the substitution of Pd (II) for Mg (II)?

    1. If yes, is it possible to synthesize and purify a Pd complex as a standard compound for comparison study?

    2. If no, compound 7 is forming another complex of Pd, what is the possible structure?

  3. For the titration of porphyrazine 7 with Pd ions, in Figure 1 UV-vis spectra, the trend in Fig 1 from 0 eq to 1 eq seems strange, why is there no clear trend? Though the authors try to explain the behavior of 0.1 of Pd(II) peaks, it is dubious. More extensive discussion is needed. Also similarly for 2 eq - 5 eq, no clear trend as well.

  4. Could the authors please provide the emission spectra of the solution of porphyrazine 7 and the solution after Pd titration?

  5. In the computational studies, the authors use HUMO LUMO bands to show agreement of excitation states in wavelength and claimed agreement of UV-vis spectra. I wonder if there is any necessity to calculate and generate UV-vis spectra?

  6. What is the novelty and specific motivation of the study synthesis in thiophene, rather than some other group? Any collective discussion with previous studies? For example, the publication of https://doi.org/10.1515/hc-2019-0001. Could the authors please elaborate with more discussions?

Other minor comments:

  1. In some parts such as line 112, the author misspells “thiophene”

Author Response

(The authors gave the same response as above.)

Round 2

Reviewer 2 Report

Comments and Suggestions for Authors

Thanks to the reviews responding to the comments.

1. In the cover letter for question #2. The authors have provided possible structure of the porphyrazine bimetallic complex. I wonder if the authors could consider discussing and adding to the manuscript or supplemental? (Probably adding to supplemental is more appropriate).

2. Continue on last comments, if the possible structure is as the authors provided. With titration with higher eq, this porphyrazine could be extended to combine with more Pd(II) ions. Could the authors provide more insights regarding it? BTW, the authors mentioned the Q band change when concentration changes from 1:2-1:4 and 1:5, does this have any relationship with the structure? How to interprete?

3. In the cover letter question #3, thanks to the authors to identify the that mistaken change of the first two spectra, 0 and 0.1 eq. Could the authors please double confirm 0.5 eq spectra as well?

Author Response

Dear Reviewer,

We have addressed all your comments stated in the second round of the revision process. Please find our response attached as a PDF file alongside the revised version of our manuscript.

Kind regards,

Tomasz Koczorowski
